# Transcriptomic–Proteomic Analysis Revealed the Regulatory Mechanism of Peanut in Response to *Fusarium oxysporum*

**DOI:** 10.3390/ijms25010619

**Published:** 2024-01-03

**Authors:** Mian Wang, Lifei Zhu, Chushu Zhang, Haixiang Zhou, Yueyi Tang, Shining Cao, Jing Chen, Jiancheng Zhang

**Affiliations:** Shandong Peanut Research Institute, Qingdao 266100, China; wmshmilyzcx@126.com (M.W.); azhulf@163.com (L.Z.); peanutzhangchushu@163.com (C.Z.); pro.zhouhaixiang@163.com (H.Z.); yueyit@126.com (Y.T.); caoshining@163.com (S.C.); cjibfc@sina.com (J.C.)

**Keywords:** peanut rot, *Fusarium oxysporum*, transcriptome analysis, proteomic profiling, regulatory mechanism

## Abstract

Peanut Fusarium rot, which is widely observed in the main peanut-producing areas in China, has become a significant factor that has limited the yield and quality in recent years. It is highly urgent and significant to clarify the regulatory mechanism of peanuts in response to *Fusarium oxysporum*. In this study, transcriptome and proteome profiling were combined to provide new insights into the molecular mechanisms of peanut stems after *F. oxysporums* infection. A total of 3746 differentially expressed genes (DEGs) and 305 differentially expressed proteins (DEPs) were screened. The upregulated DEGs and DEPs were primarily enriched in flavonoid biosynthesis, circadian rhythm-plant, and plant–pathogen interaction pathways. Then, qRT-PCR analysis revealed that the expression levels of phenylalanine ammonia-lyase (PAL), chalcone isomerase (CHI), and cinnamic acid-4-hydroxylase (C4H) genes increased after *F. oxysporums* infection. Moreover, the expressions of these genes varied in different peanut tissues. All the results revealed that many metabolic pathways in peanut were activated by improving key gene expressions and the contents of key enzymes, which play critical roles in preventing fungi infection. Importantly, this research provides the foundation of biological and chemical analysis for peanut disease resistance mechanisms.

## 1. Introduction

*Fusarium oxysporum* (*F. oxysporum*) is a soilborne pathogenic fungus that is found throughout the world and can cause severe blights on plants by infecting their vascular bundles [1]. Indeed, the reduction in yield caused by *F. oxysporum* can be as high as 50%, and even result in no harvest [2]. Therefore, it is a severe threat to various plants, including key economic and horticultural crops, such as rice (*Oryza sativa*), tomato (*Solanum lycopersicum*), potato (*Solanum tuberosum*), and peanut (*Arachis hypogea*) [3,4,5]. The induction of pod, root, and stem rot by *F. oxysporum* in peanuts has been widely observed in peanut production areas in China, including Hebei, Henan, Shandong, Guangdong, Jilin, and Hainan Province, and this disease has become a significant factor that limits the yield and quality of peanut [6,7].

The infection of plants with *F. oxysporum* induces a series of reactions: immune response, secretion of various compounds, and increase in enzyme activity [8,9]. For example, the proteins associated with the photosynthetic apparatus are extensively inhibited, while energy metabolism/protein synthesis and turnover are usually upregulated. This indicates a major redirection of cellular metabolism [10,11]. Other common features include the regulation of glucose metabolism; cell walls; reactive oxygen species (ROS); pathogenesis-related (PR) protein expression; and secondary metabolites, such as flavonoids and polyphenolics [12]. After infection with *F. oxysporum*, the levels of expression of cinnamoyl-CoA reductase (CCR-1), a simple NLR gene (RPP8), and chitinase genes are highly upregulated in castor beans (*Ricinus communis*) [13]. The upregulation of CCR-1 involved in lignin biosynthesis helps to neutralize the ROS in cells by increasing the activity of Superoxide dismutase (SOD) [14]. However, owing to the lack of resistant germplasm in peanut, the regulatory mechanism in peanut’s response to *F. oxysporum* remains unclear.

Studies on peanut and Fusarium mainly focused on the isolation and identification of Fusarium and its antagonistic bacteria in soil, such as plant-growth-promoting rhizobacteria [15]. An active fermentation liquor of *Serratia marcescens* (RZ-21) treatment significantly increased peanut yield, vine dry weight, root nodules, and taproot length by enhancing the activities of soil urease, sucrase, and hydrogen peroxidase at various stages [16]. A multiplex PCR detection system to detect *F. oxysporum* of peanut southern blight, stem rot, and root rot was set up [17]. Nevertheless, there are no effective measures for the treatment of peanut disease induced by *F. oxysporum*. In particular, the effects of *F. oxysporum* on the key genes and enzymes in the regulatory mechanism have not been reported.

In particular, combined multi-omics analytical methods are widely used in the study of plant–microbial interactions [18,19]. In terms of the functional research related to peanuts, transcriptome and proteome sequencing were used to assist in screening the key genes involved in aerial and subterranean pods [20], drought resistance [21,22], development [23,24], environmental stress [25], and interaction with *Aspergillus flavus* [26]. In this study, peanut stems infected with *F. oxysporum* were used as experimental materials. Transcriptome analysis was used to mine the changes in the expression of related genes. Proteome sequencing was performed to analyze the profile of protein expression. Moreover, the changes in the levels of expression of key genes and the characteristics of their expression in the tissues were verified using real-time quantitative reverse transcription PCR (qRT-PCR). According to these results, the contents of five metabolites and the activities of four key enzymes in peanut after inoculation with *F. oxysporum* were measured to reveal the physiological and biochemical reactions of peanut infected with *F. oxysporum*, which laid a theoretical foundation to elucidate the role of key metabolic pathways in peanut disease resistance.

## 2. Results

### 2.1. Histomorphological Changes in Peanut after Being Infected with F. oxysporum

When a peanut pod rots in the field, the ground tissue is intact, but the belowground pod shell rots, along with the seeds in serious cases, which results in severe yield reduction or even the loss of an entire harvest (Figure 1A,B). The stems and leaves of peanut were inoculated with *F. oxysporum.* Within 3 days after inoculation, different degrees of rot appeared on the inoculated points. The lesion areas with about 1 mm diameter turned yellow-brown and began to spread to the surrounding tissues. Within 7 days after inoculation, the inoculation sites and the surrounding tissues were seriously decomposed, where the dark brown lesions expanded to about 3 mm and continued to widely spread (Figure 1C–G). A total of 300 peanut varieties were infected with the pathogen for disease resistance, and some of the results are shown in Figure 1H–K. The sizes and colors of the lesions and the incidence rates were selected as screening indexes where the fresh seeds of the peanuts could be infected with the pathogen; the high incidence rates indicated a serious lack of disease resistance in these peanut varieties.

### 2.2. Results of Transcriptome Sequencing

To identify the differentially expressed genes (DEGs) in peanut stems before and after infection with *F. oxysporum*, 7 d after inoculation, the stems were used as the materials for transcriptomic analysis. The Illumina Nova Seq 6000 sequencing platform was used on six samples. The transcriptome data of six samples were uploaded to BioProject (PRJNA1020986, TaxID: 226205). A total of 67.48 GB of clean data were obtained, and the clean data of each sample reached 10.81 GB. The percentage of the Q30 base was ≥93.49%. When the effective data after quality control were aligned with the peanut reference genome, the alignment efficiency ranged from 93.68% to 94.57% (Table 1). Finally, 6072 novel genes were identified, and 4959 of them were functionally annotated. Among them, 519, 1465, 784, and 1772 genes were annotated in the Clusters of Orthologous Genes (COG), Gene Ontology (GO), the Kyoto Encyclopedia of Genes and Genomes (KEGG) and EuKaryotic Orthologous Groups (KOG) databases, respectively. To assess the consistency between the biological replicates, all the samples were clustered hierarchically based on the correlation coefficient γ2 between each sample. The results show that the control and inoculated samples were clustered separately, whereas the three biological replicates of each group were clustered together, indicating that the biological replicates of each group were highly similar (Figure 2).

### 2.3. Analysis of the DEGs

To identify the relevant genes in response to fungal infection, differential expression analysis was performed. Using fold change ≥ 2 and FDR < 0.01 as the screening criteria, 9451 DEGs were obtained, involving 5894 upregulated genes and 3557 downregulated genes. Meanwhile, when fold change ≥ 4 and FDR < 0.001 were used as the screening criteria, a total of 3746 DEGs were screened, involving 2758 upregulated genes and 988 downregulated genes (Appendix A). Among the DEGs, 1496, 1939, and 1303 genes were annotated in the COG, GO, and KEGG databases, respectively. The GO enrichment analysis showed that the DEGs were primarily enriched in the metabolic process, cell signal transduction process, and stress response (Appendix A).

To identify the signal transduction pathways, KEGG enrichment analysis of the DEGs was performed, which indicated that the DEGs were primarily enriched in circadian rhythm-plant (ko04712), plant–pathogen interaction (ko04626), and flavonoid biosynthesis (ko00941) pathways (Appendix A), whereas the upregulated genes were primarily enriched in flavonoid biosynthesis (ko00941), circadian rhythm-plant (ko04712), and plant–pathogen interaction (ko04626) pathways (Figure 3A) and the downregulated genes were primarily enriched in photosynthesis (ko00195), photosynthesis-antenna proteins (ko00196), and carbon fixation in photosynthetic organisms (ko00710) pathways (Figure 3B). This indicates that peanuts activate many genes in plant disease resistance or secondary metabolism pathways during the process of resistance to pathogen infection.

### 2.4. Proteome Sequencing

The proteome data of six samples were uploaded to the Proteome Dataset (https://www.ebi.ac.uk/pride/archive/, accessed on 5 December 2023). A total of 50,277 peptide proteome sequences were obtained, and 30,615 of them were identified as specific peptides. A total of 7524 proteins were annotated, and 4061 of them contained quantitative information. As indicated, three analytical methods were used to identify the reproducibility of the protein data, which indicated that the three replicate samples were highly reproducible (Figure 4A–C). Compared with CK, 305 differentially expressed proteins (DEPs) were found in the LG samples, involving 157 upregulated and 148 downregulated proteins. The list of DEPs is shown in Appendix A, and the scatter diagram of the DEPs is shown in Figure 4D.

### 2.5. Analysis of the DEPs

To analyze the mechanism of the response of peanut resistance to pod rot, a GO enrichment analysis of the DEPs was performed (Appendix A). For example, the upregulated DEPs were primarily enriched in the cytosol (GO:0005829) pathway related to cell composition; oxidoreductase activity (GO:0016491), cofactor binding (GO:0048037), coenzyme binding (GO:0050662) pathways associated with molecular function; and isoprenoid biosynthetic process (GO:0008299), isoprenoid metabolic process (GO:0006720), cellular hormone metabolic process (GO:0034754) pathways related to biological process directions.

A KEGG enrichment analysis showed that the upregulated DEPs were primarily enriched in the Terpenoid backbone biosynthesis (aip00900), alpha-Linolenic acid metabolism (aip00592), and Flavonoid biosynthesis (aip00941) pathways (Figure 5A, Appendix A). The downregulated DEPs were primarily enriched in the photosynthesis (aip00195), Carbon fixation in photosynthetic organisms (aip00710), and photosynthesis-antenna proteins (aip00196) pathways (Figure 5B, Appendix A). An additional analysis showed that nine upregulated and seven downregulated proteins were annotated as being related to the phenylpropanoid metabolic pathway (Table 2). A protein–protein interaction analysis showed that there may be interactions between the 1WY37S.1, LGAM8W.1, 0BML12.1, 5H4H17.1, 1KSV8R.1, SGZ2CH.1, and P6MJUK.1 proteins in the phenylpropanoid metabolic pathway (Appendix A). These results indicate that peanuts activated a large number of metabolic pathways and produced many disease-resistant compounds to resist the pathogenic infection.

### 2.6. Correlation Analysis of the Transcriptome and Proteome Sequencing

To clarify the co-expression of peanuts at the protein and transcript levels during infection by *F. oxysporum*, the quantitative transcriptome and proteome data were jointly analyzed. The correlation R-value between the two sets of omics data was 0.58, as shown by the scatter plot (Figure 6A). Sample clustering heatmap and principal component analysis (PCA) (Appendix A) showed that the two sets of omics data were highly reproducible and produced positive correlations between the proteome and the transcript among duplicate samples (Appendix A).

To compare the one-to-one correspondence between differential proteins and transcripts, the intersection of the two sets of data was analyzed, and the expression of transcripts and proteins was divided into six different expression profiles. The Venn diagram showed that the numbers of DEPs in six regulatory types (up–up, up–unchanged, unchanged–up, unchanged–down, down–unchanged, and down–down) were 73, 2685, 84, 127, 967, and 21, respectively (Figure 6B). The top 10 DEPs in both up–up type and down–down type are shown in Table 3. A co-enrichment analysis of the GO classification functions (Appendix A) and a KEGG enrichment analysis were performed using a clustering heatmap (Figure 6C, Appendix A). Among these DEPs, 17 upregulated and 5 downregulated DEPs were enriched in the phenylpropanoid metabolic pathway. This implies that in response to pathogen infection, peanuts can activate the expression of key genes linked to plant disease resistance and secondary metabolism pathways. Moreover, pathogen invasion has the potential to disrupt or obstruct the photosynthesis pathway in plants, thereby enhancing the success of their pathogenicity.

### 2.7. Verification of the Levels of Expression of Key Genes Using qRT-PCR

Plant metabolic pathways participate in plant disease resistance by producing multiple secondary metabolites, such as lignin, flavonoids, and alkaloids [27,28]. Phenylalanine ammonia lyase (PAL) and 4-coenzyme A ligase (4CL) are key enzymes in metabolic pathways that enhance plant resistance to pathogen infection [29,30]. According to the enrichment of KEGG signaling pathways, 11 DEGs that encoded PAL, chalcone isomerase (CHI), and so on, were selected for the analysis and verification of expression. The expression levels of *1WY37S.1*, *LGAM8W.1*, *0BML12.1*, *5H4H17.1*, *1KSV8R.1*, *SGZ2CH.1*, and *P6MJUK.1* continued to increase as the time of inoculation increased, while those of the *ULUR0X.1*, *333C3Q.1*, *JB63H4.1*, and *P96X61.2* genes were downregulated (Figure 7). The changes in the expressions of these 11 genes were consistent with the transcriptome test, thereby validating the reliability of the transcriptomic results.

### 2.8. Expression of the Key Genes in the Phenylpropanoid Metabolic Pathways

To clarify the specificity of the expression levels of these key 11 genes, we examined them in the roots, stems, leaves, and cotyledons using qRT-PCR (Figure 8). The results showed that *1WY37S.1*, *LGAM8W.1*, *0BML12.1*, *5H4H17.1*, *1KSV8R.1*, *SGZ2CH.1*, and *ULUR0X.1* were highly expressed in the roots. *0BML12.1*, *SGZ2CH.1*, *5H4H17.1*, *P6MJUK.1*, and *P96X61.2* were highly expressed in the leaves; *333C3Q.1*, *JB63H4.1*, *P6MJUK.1*, and *P96X61.2* were highly expressed in the cotyledons; while *JB63H4.1*, *P96X61.2* and *333C3Q.1* were highly expressed in the stems. Among them, 1WY37S.1 encodes a cytochrome P450 family protein, 5H4H17.1 encodes PAL, LGAM8W.1 encodes CHI, SGZ2CH.1 encodes cinnamic acid-4-hydroxylase (C4H), P96X61.2 encodes fructose-diphosphate aldolase, and P6MJUK.1 encodes fructose-1,6-bisphosphatase. These genes showed specificity in peanut tissue. These data indicate that the expression levels of these genes that varied in different peanut tissues might play roles in various tissue sites.

### 2.9. Determination of the Activities of Key Enzymes

To analyze the intrinsic changes of peanuts after *F. oxysporum* infection, the contents of four key enzymes were measured. It was revealed that the PAL activities of peanut increased by 35% within 2 d after inoculation, but with no significant change later (Figure 9B). The SOD activity increased by 77% with the extension of inoculation time (Figure 9C). The activity of 4CL increased by 66% at 2 d after inoculation; however, it decreased at approximately 5 d after inoculation (Figure 9A). The overall trend of C4H activity decreased by 32% (Figure 9D). These results indicate that the inoculation of peanut with *F. oxysporum* activated some key enzymes and promoted the formation of metabolites to induce resistance against the pathogen.

### 2.10. Measurement of the Contents of Metabolites in Peanut

The function of many metabolites in plant disease resistance has been clearly defined, and the contents of these metabolites differ significantly in different types of germplasm [5,31,32,33,34]. In order to determine the changes in metabolites in peanuts infected with *F. oxysporum*, we measured the changes in the contents of some specific secondary metabolism products (Figure 9E–I). The contents of total flavonoids (TF) and malondialdehyde (MDA) increased by 50% and 39%, respectively, with the extension of inoculation time. The total plant phenolics (TP) decreased first and then increased by 23%. The content of flavonoids decreased by 33% with the extension of inoculation time. The content of soluble sugar noticeably increased by 57%. In summary, the contents of phenylpropanoid and flavonoid metabolites in peanut changed significantly following infection, suggesting that the secondary metabolic pathways may be involved in the response against pathogenic fungi infection.

## 3. Discussion

This study focused on the mechanism of interaction between peanut and *F. oxysporum*. Currently, there is a lack of reported histomorphological changes in peanuts infected with Fusarium. According to the laboratory inoculation results of peanuts at different growth stages and different tissues, *F. oxysporum* can infect all the tissues of peanut. The results from germplasm screening indicated a serious shortage of resistant peanut varieties. Therefore, studying and clarifying the mechanisms underlying the interaction between peanuts and Fusarium provides an important theoretical basis for enhancing peanut resistance against this pathogen.

In terms of plant–Fusarium interactions, the transcriptome and proteome analysis revealed important results. Particularly, quantitative proteome and transcriptome abundance maps of 15 major sweet cherry (*Prunus avium*) tissues yielded 29,247 genes and 7584 proteins [35]. A total of 1850 genes and 356 protein species were differentially regulated in *Cucurbita ficifolia Bouché* (Cucurbitaceae) leaves 2 and 4 d after *F. oxysporum* inoculation [36]. In peanut research, omics analysis has been widely applied in various aspects. For instance, in investigating the response mechanism of peanuts to low-temperature stress during imbibition, 5029 proteins were identified and quantified; among them, 104 proteins were found to be differentially expressed [37]. However, there are no reports on combined omics analysis elucidating the mechanisms involved in the interaction between peanuts and Fusarium. Here, a total of 3746 DEGs and 305 DEPs were found. The patterns of expressions of 11 key genes were identified using real-time PCR and were consistent with the results of the transcriptome sequencing. This indicates that the sequencing results of this study were reliable, and key metabolic pathways and target genes were identified, which provides reliable data support for subsequent research. Based on these studies, integrating multi-omics data and gene regulatory network inference to describe signaling pathways and discover new regulations can accelerate the discovery of key genes and key proteins.

Studies of omics research have shown that the interaction between plants and pathogens can initiate many reactions, such as the immune response, increase in metabolites, increase in antioxidant enzyme activity, and expression of disease resistance genes [38]. For example, maize (*Zea mays*) was infected with verticillium wilt, the DEGs and DEPs were primarily enriched in glutathione metabolism, starch and sucrose metabolism, amino sugar and nucleotide sugar glycometabolism, linoleic acid metabolism, and phenylpropanoid biosynthesis [39]. In a study of soybean (*Glycine max*) root rot disease, key pathways and DEPs were found to be involved in phenylalanine metabolism, plant hormone signal transduction, and plant–pathogen interaction, while PAL, calcium-dependent protein kinase (CPK), and other defense-related proteins were upregulated [40]. Similarly, the upregulated genes and proteins in peanut after *F. oxysporum* inoculation were primarily involved in plant–microbe interaction, flavonoid synthesis, and circadian rhythm. The downregulated genes and proteins were primarily enriched in photosynthesis pathway and its related pathways. The interaction mechanism between peanut and Fusarium wilt is similar to that between soybean and root rot, but different from that between maize and verticillium wilt. Both of them upregulated the expression of key genes, such as PAL, and activated secondary metabolism-related pathways. This suggests that peanut, like other leguminous plants, defends against fungal infection by promoting the expression of key metabolic pathways and limiting photosynthesis-related pathways after fungal infection.

Furthermore, the upregulation of disease-resistance-related genes can enhance plant immunity [40,41,42]. Similar to the mechanism observed in potato–Fusarium interaction, we examined the expression levels of 11 relevant genes. Consistent with previous studies, the expression levels of several key enzyme genes (PAL, 4CL, C4H, and CHI) were found to be enhanced upon infection [30]. Peanut may promote the production of metabolites to resist *F. oxysporum* infection by upregulating the expression of rate-limiting enzyme genes in metabolic pathways. Interestingly, among these genes, the upregulated genes were highly expressed in the roots, while the downregulated genes were highly expressed in the stems. This is a curious result, which is probably similar to the characteristics of peanut flowering above ground and pod-bearing below ground, and its gene expression is very tissue-specific. These results indicate that these genes may play specific roles in different peanut tissues. The results of PAL gene expression in different tissues were contrary to the results showing that *Fagopyrum tataricum* PAL gene transcript levels were higher in flowers and seeds than in roots, stems, and leaves [43].

The accumulation of certain metabolites during the plant–pathogen interaction was shown to improve disease resistance [13,31,41,43]. For instance, terpenoids can accumulate during chrysanthemum–Fusarium interaction [44], while lignin accumulation promotes alfalfa resistance against Fusarium [31]. In this study, the activities of PAL and 4CL increased significantly 2 d after inoculation, indicating that the inoculation of *F. oxysporum* promoted the process of phenylpropanoid metabolism pathways, promoted the metabolism of some phenolics, and increased the content of TF to fight against the disease. This is also similar to the determination related to the disease course of rice blast [45,46]. However, the results of decreased C4H activity are different from other reports [47]. The results demonstrated that peanut infected with *F. oxysporum* produced a large amount of sorting compounds to improve the resistance of the plant to disease.

More interesting is that the peanut phenylpropanoid metabolic pathways also showed a remarkable role in the study of peanut nutrient metabolism [20], drought [21], and needle and pod development [23,48]. Here, some upregulated differentially expressed transcripts and proteins accumulated in the flavonoid metabolic pathways. However, the decrease in the contents of flavonoids as part of the total flavonoid contents indicated that the metabolic pathways of flavonoids were initiated. These results indicate that peanut may selectively promote the process of some metabolic pathways during the process of resisting *F. oxysporum* infection (Figure 10). Other physiological and biochemical indices showed an increased soluble reducing sugar and MDA contents, and the continuously increased SOD activity improved peanut’s immunity by regulating glycometabolism and antioxidant pathways when resisting pathogen infection.

In light of the frequent occurrence of peanut Fusarium disease and the limited availability of resistant varieties, it is imperative to investigate the mechanisms underlying peanut–Fusarium interaction. This study represents an initial step toward understanding these mechanisms. Currently, our focus lies in breeding Fusarium-resistant peanuts by screening existing germplasm resources for disease-resistant traits. Additionally, EMS and radiation methods were utilized to generate mutant materials with enhanced resistance. Subsequently, conventional crossbreeding will be employed using these resistant germplasms to establish an isolated population. These studies can contribute to the improvement of peanut disease resistance.

## 4. Materials and Methods

### 4.1. Plants

A previous field experiment showed that the peanut cultivar “Huayu 20” was more resistant to pod rot in the field than other cultivars [7]. Therefore, it was selected as the test variety for this experiment. The peanut seeds used in this study were stored in our laboratory at 4 °C in a refrigerator. The peanut seedlings were cultured at 25 °C in 16 h light/8 h dark. The roots, stems, leaves, and cotyledons from the peanut seedlings were collected after 2 weeks, frozen in liquid nitrogen, and then stored at −80 °C in an ultracold freezer.

### 4.2. Strains

An isolate of *F. oxysporum*, strain PPRF-05, was isolated, purified, and preserved. This isolate was deposited in the China General Microbiological Culture Collection Center under no. 18130.

### 4.3. Treatment

In the laboratory, peanut seeds sterilized with 70% anhydrous ethanol were used for hydroponic cultivation. The mycelia of *F. oxysporum* that had been cultured in Petri dishes at 37 °C in the dark in an incubator for 15 d were scraped off. A volume of 10 mL PDA was added, shaken vigorously, and filtered, and a spore solution at a concentration of 1 × 10^6^ was prepared. The 4-week-old seedlings were treated with the spore solution, and tissue samples were taken at 0, 2, 5, and 7 d after inoculation. The RNA from plant tissues was extracted and transcribed into cDNA for storage.

Peanut seedlings were inoculated with half-sections of well-cultured *F. oxysporum* and were seeded upside down on the lower end of the stem, and samples were collected at 3 and 7 d after inoculation. The tissue at the middle end of the stem was labeled as LG. The control groups were not inoculated with the fungus and were labeled as CK. Two samples were established with three biological replicates and divided into three parts for the transcriptome and proteome analysis. One part was used to extract the RNA and reverse transcribe it into cDNA for qRT-PCR to access the gene expression profile. Another part was sent to Beijing Biomarker Technologies Co., Ltd. (BMK, Beijing, China), for the transcriptome analysis, and the third part was sent to Hangzhou Pioneer Proteomices Co., Ltd. (Hangzhou, China), for proteome analysis.

### 4.4. Analysis of Transcriptome Sequencing

An Illumina high-throughput sequencing platform (Illumina, San Diego, CA, USA) was used to sequence the cDNA library, and a large number of high-quality data was generated. Base quality and mismatch analyses were performed on these data. The HISAT2 system was employed using the peanut genome sequence as the reference genome (Tifrunner.gnm1. ann1.CCJH: https://peanutbase.org/data/, accessed on 15 February 2018), and StringTie was used for the assembly, splicing, quantitative analysis, and new gene annotation. BLAST was used to perform sequence alignments with the NR, SwissProt, GO, COG, KOG, and KEGG databases, whereas the KEGG Orthology results of the new genes were acquired from KOBAS2.0. After predicting the amino acid sequence of the new gene, HMMER was used to compare the sequence with the Pfam database to obtain the annotation information of the new gene. The fragments per kilobase of transcript per million mapped reads (FPKM) were used as a measure of the levels of transcript or gene expression. Pearson’s correlation coefficient was used as an index to evaluate the correlation of biological replicates. The closer the R2 was to 1, the stronger the correlation between the two replicates is. DESeq was used to analyze the differential expression between the sample groups, and the data in different samples were used to screen the differentially expressed genes (DEGs). Genes with a false discovery rate (FDR) ≤ 0.001 and log2FC ≥ 4 were defined as DEGs.

### 4.5. Analysis of Proteome Sequencing

The extracted peanut stem proteins were digested by enzymes to obtain the peptide fragments, which were analyzed using liquid chromatography–mass spectrometry (NanoElute, Bruker Daltonics Inc., Beijing, China). Maxquant (v1.6.6.0) was used to retrieve the secondary mass spectrometry data. The retrieval parameter settings were as follows: the database was UniPort (29,947 sequences), the anti-library was added to calculate the FDR caused by random matching, and the common contamination library was added to the database to eliminate the influence of contaminating proteins in the identification results. For the quality control detection, multi-mass spectrometry data were used. A bioinformatics analysis was performed on the data that met the analytical requirements using the following software and platforms. To fully understand the identified and quantified proteins in the data, the functions and characteristics of these proteins were annotated in detail in terms of the GO and KEGG pathways and the COG functional classification.

For the annotation of all the proteins identified and the screening of DEPs, the DEPs in each comparison group were subjected to the GO classification and KEGG pathway, respectively. The *p*-value obtained using Fisher’s exact test showed the functional categories and pathways with significant enrichment (*p* < 0.05) of the differential proteins. According to their fold change, they were divided into four fractions designated Q1 to Q4. For each Q group, GO and KEGG classifications and protein domain enrichment were performed. Furthermore, the cluster analysis was performed to find the functional correlation of proteins with different fold changes. After comparison with the STRING (v.10.5) protein network interaction database, the differential protein interaction relationship was assessed based on the confidence score > 0.7, and the interaction network of the DEPs was visualized using the R package (4.3.2) “networkD3” tool.

### 4.6. DEPs and Transcript Screening

Quantitative experiments were performed in triplicate at the transcriptome and proteome levels to evaluate the quantitative repeatability of the two omics and the correlation of the sample’s expression at the two omics levels. A Pearson correlation coefficient and PCA were used to compare the quantitative reproducibility of the two omics. In addition, by comparing the quantitative correlation of the two omics, the potential regulatory relationships between the proteins and transcripts were identified and shown using a scatter plot and Venn diagram.

The quantitative transcriptome results were analyzed via a difference analysis to obtain the log2FC and *p*-value of different statistical tests for the relative expression of the transcriptome in each comparison group. Because of the large amount of data used for the transcriptome differential analysis, a Benjamin–Hochberg (B-H) test was performed on the *p*-value to further reduce the FDR. A significantly upregulated transcript was defined as log2FC > 2 and a verified *p*-value < 0.001, while the downregulated transcript was defined as log2FC < −2 and a verified *p*-value < 0.001. Similarly, the ratio value of the relative proteome expression in each comparison group and the difference statistical test *p*-values were obtained via a difference analysis. The amount of data of the DEPs was smaller than that of the transcriptome; thus, further FDR analysis on these *p*-values was not required. If the ratio was >2 and *p* < 0.05, the protein was considered significantly upregulated; and if the ratio was <0.5 and *p* < 0.05, the protein was considered significantly downregulated.

### 4.7. Verification Using qRT-PCR

qRT-PCR was conducted using an SYBR Green Premix Pro Taq HS qPCR Kit (Thermo Fisher Scientific, Waltham, MA, USA). The total RNA was extracted from the roots, stems, leaves, and cotyledons of the flowers using a TaKaRa Mini BEST Plant RNA Extraction Kit, and reverse transcription of the microRNAs (miRNAs) was performed using Mir-X miRNA First-Strand Synthesis (Takara Bio, Dalian, China). The reverse-transcribed cDNA was used as a template and normalized with the internal reference 5.8s rRNA. The primer sequences were designed using Primer 5.0 software (Table 4) and synthesized using Sangon Bioengineering Co., LTD (Shanghai, China).

GraphPad Prism 5 (San Jose, CA, USA) was used for the data of the qRT-PCR. The 2-∆∆CT method was used to calculate the relative expression level. After a correction using one-way ANOVA analysis, the data was analyzed using the Newman–Keuls method. A *p*-value < 0.05 meant a significant difference. *Ahactin11* was used as the internal reference gene to calculate the relative expression level. The expression level of the CK was taken as “1”, and that of the treated sample was calculated and plotted. Each result is the mean ± standard deviation of 3 repetitions.

### 4.8. Determination of Five Metabolites and Four Enzymes from Peanut Data

The stems of well-grown peanut seedlings were sampled at 0, 2, 5, and 7 d after inoculation with *F. oxysporum*. Three biological replications were established, and the samples were sent to MDBio, Inc. (Taipei, China) to determine the activities of PAL, SOD, C4H, and 4CL, and the contents of MDA, plant flavonoids, TP, TF, and soluble sugars.

### 4.9. Data Processing

When analyzing the changing trend of the contents of 9 metabolites over 0, 2, 5, and 7 days, all data were statistically analyzed and plotted using GraphPad Prism 5 (San Jose, CA, USA) software. After correction using one-way ANOVA analysis, the data were analyzed using the Newman–Keuls method. A *p*-value < 0.05 meant a significant difference. The test value of day 0 was used as the control to calculate the increase or decrease at 2, 5, and 7 days after inoculation. The results of 3 repetitions are shown as the mean ± standard deviation.

## 5. Conclusions

According to the above results, we speculated that peanut, like other plants, had a similar response to *F. oxysporum*. The expressions of genes (PAL, 4CL) in peanut related to immune response were stimulated after being infected with *F. oxysporum*. The synthesis of key enzymes (SOD, PAL, 4CH, CHI) was promoted, and the biosynthesis of metabolites, such as polyphenols and flavonoids (MDA, soluble sugars, TF, TP) was involved, which promoted the production of metabolites to improve the ability of autoimmunity (Figure 10). These findings provide new insights into the molecular mechanisms of peanut’s response to *F. oxysporum*. Importantly, this research provides the foundation of biological and chemical analysis for the breeding of resistant varieties.

## Figures and Tables

**Figure 1 ijms-25-00619-f001:**
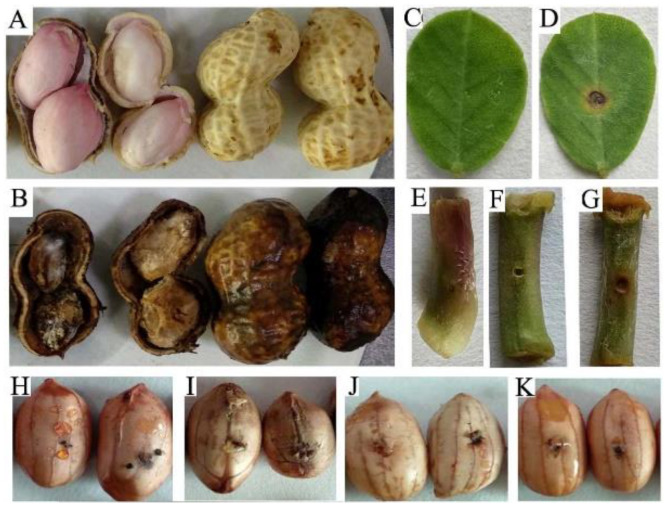
Peanut phenotype infected with *F. oxysporum*. (**A**) Field peanut fresh pods and seeds. (**B**) Rotten peanut pods and kernels collected from the field. (**C**) Control samples of peanut seedling leaves that were not inoculated with *F. oxysporum*. (**D**) Peanut leaves infected with *F. oxysporum*. (**E**) Control samples of peanut stem tissues. (**F**,**G**) Peanut stem tissue infected with *F. oxysporum* for 3 days and 7 days, respectively. (**H**–**K**) Four peanut germplasm resources (Z-70, Z-29, Z-102, and Z-128, respectively) stored in our laboratory infected with *F. oxysporum*.

**Figure 2 ijms-25-00619-f002:**
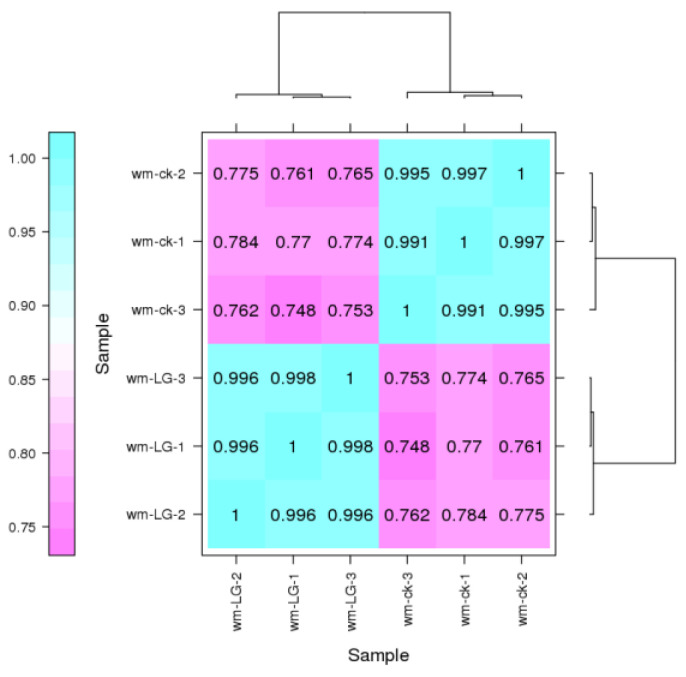
Expression calorimetry of correlation between two samples. Pearson’s correlation coefficient was used as an index to evaluate the biological reproducibility. The closer R2 is to 1, the stronger the correlation between the two samples is. The CK and LG samples had three biological duplications.

**Figure 3 ijms-25-00619-f003:**
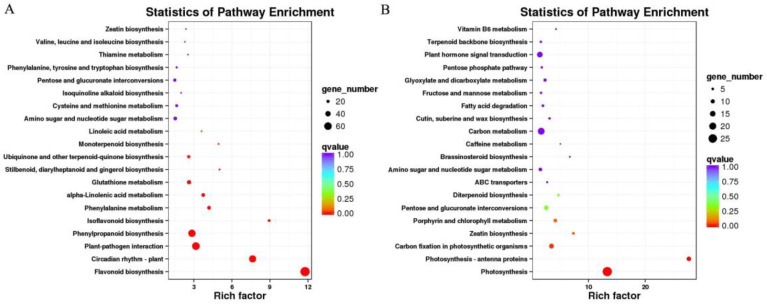
Plot of KEGG pathway enrichment of differentially expressed genes. Each circle in the graph represents a KEGG pathway, with the name of the pathway in vertical coordinates and the enrichment factor in horizontal coordinates, which represents the ratio of the proportion of genes annotated to a pathway in a differential gene to the proportion of genes annotated to that pathway in all genes. The higher the enrichment factor, the more significant the enrichment level of differentially expressed genes in this pathway. The color of the circle represents Q value, and Q value is *p*-value after correction of multiple hypothesis test. The smaller Q value is, the more reliable the enrichment significance of differentially expressed genes in this pathway is. The size of the circle indicates the number of genes enriched in the pathway, and the larger the circle, the more genes. (**A**) is the enriched scatter plot of upregulated genes and (**B**) is the scatter plot of downregulated genes.

**Figure 4 ijms-25-00619-f004:**
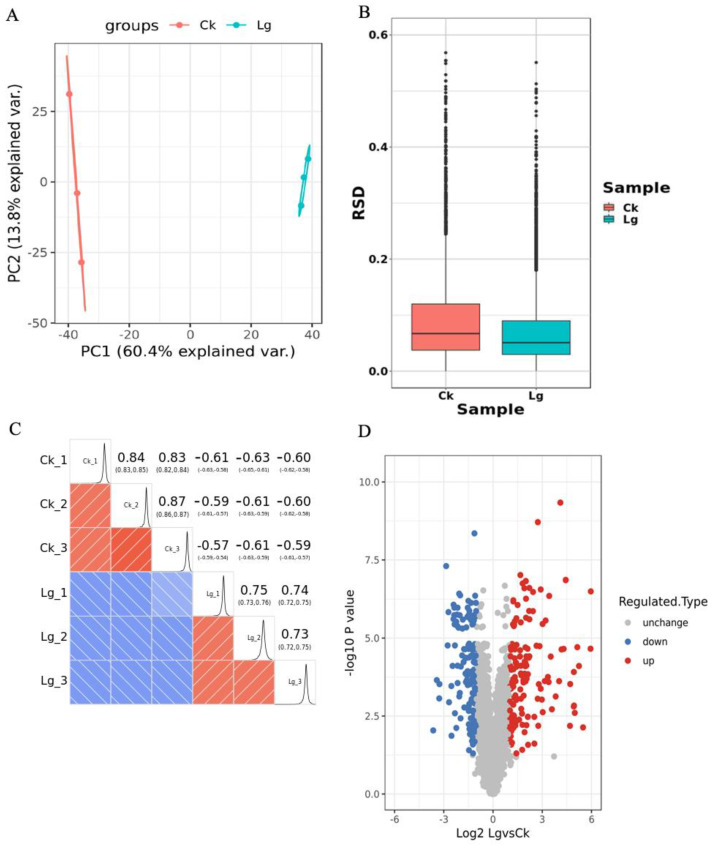
Analysis of protein data reproducibility and scatter plot of differential expression proteins. (**A**) Quantitative principal component analysis scatter plot of protein between repeated samples. (**B**) A boxplot of quantitative RSD distribution of proteins between repeated samples. (**C**) Heatmap of Plzeň correlation coefficient for protein quantification between two samples. (**D**) Differential expression protein quantitative volcano plot. Horizontal axis for protein expression log2 value. Vertical axis for protein expression −log10 *p*-value.

**Figure 5 ijms-25-00619-f005:**
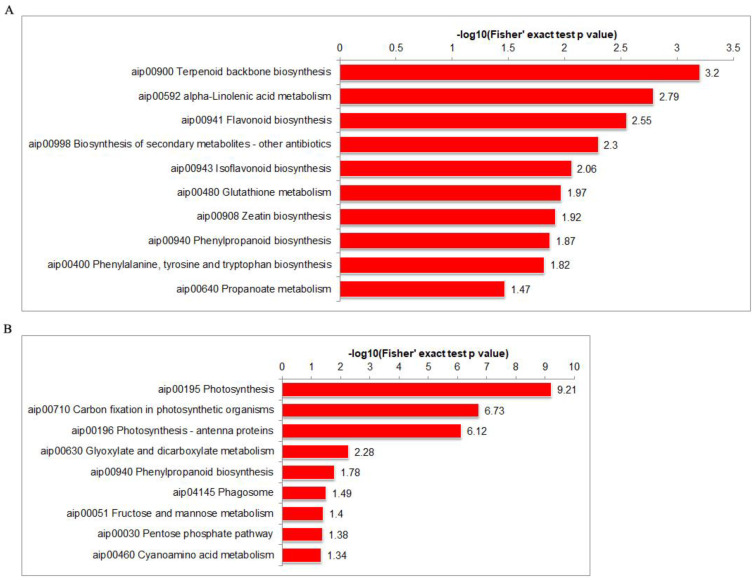
A KEGG enrichment analysis of DEPs. The vertical axis is protein-enriched KEGG pathway and the horizontal axis is protein *p*-value in −log 10. (**A**) The pathway of upregulated DEPs enrichment. (**B**) The pathway of downregulated DEPs enrichment.

**Figure 6 ijms-25-00619-f006:**
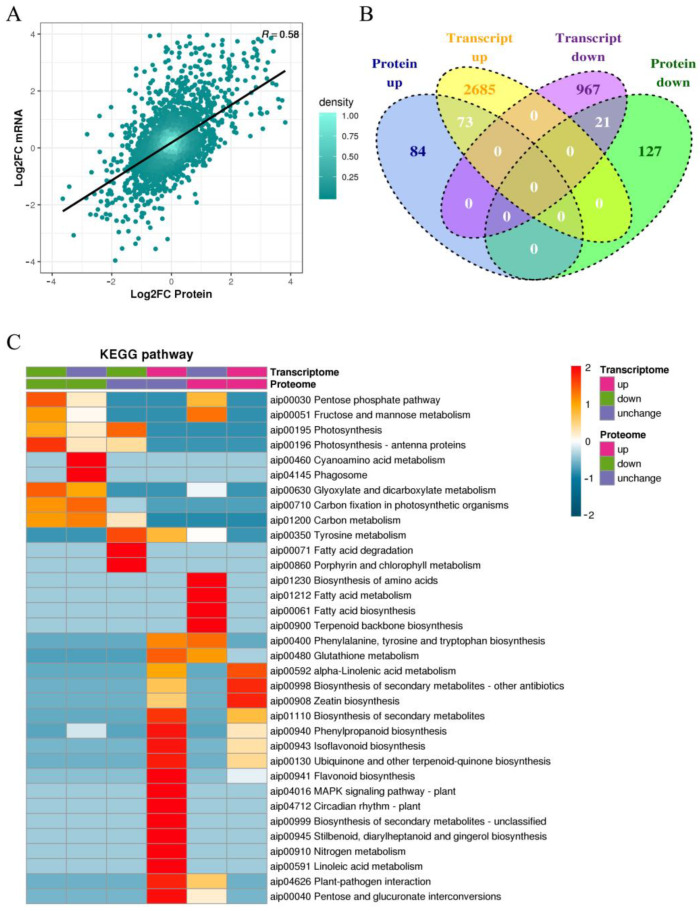
Correlation analysis of the transcriptome and proteome sequencing. (**A**) The scatter plot of the transcript and its corresponding protein expression. The vertical axis is DEG *p*-value in log2FC and the horizontal axis is DEP *p*-value in log2FC. (**B**) The Venn diagram of DEPs and DEGs in these six regulatory types. (**C**) Cluster heatmap of DEPs and DEGs enriched in KEGG pathway in six regulatory relationships.

**Figure 7 ijms-25-00619-f007:**
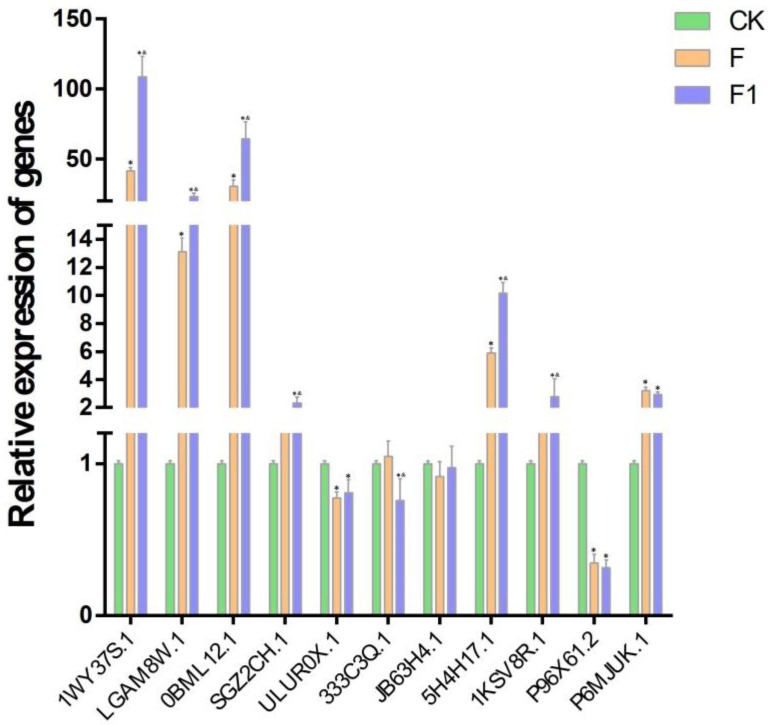
Expression characteristics of 11 genes in phenylpropane metabolism pathway. Horizontal coordinates: peanut genes. Vertical coordinates: gene expression fold. CK: control sample not inoculated with *F. oxysporum*. F: sample inoculated with *F. oxysporum* for 3 days. F1: sample inoculated with *F. oxysporum* for 7 days. Each error bar represents the mean ± standard deviation of 3 repetitions. * means *p*-value < 0.05 for CK vs. F/F1. & means *p*-value < 0.05 for F vs. F1.

**Figure 8 ijms-25-00619-f008:**
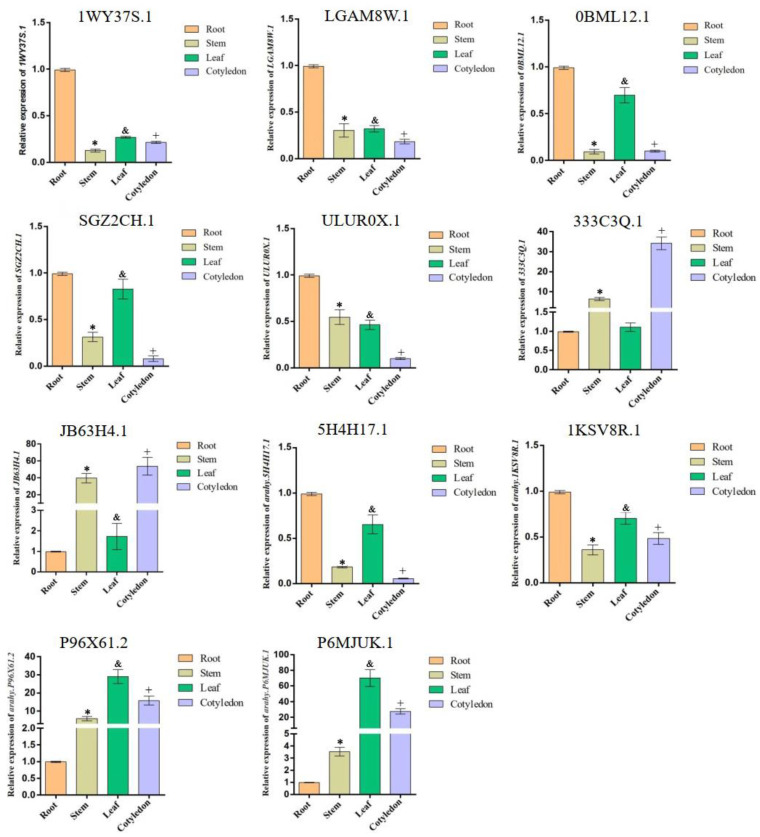
Tissue expression specificity analysis of 11 genes. Horizontal coordinates: different peanut tissues, with the root marked by orange, stem marked by yellow, leaf marked by green, and cotyledon marked by purple. Vertical coordinates: relative gene expression level. Each error bar represents the mean ± standard deviation of 3 repetitions. * means *p*-value < 0.05 for stem vs. root. & means *p*-value < 0.05 for leaf vs. root. + means *p*-value < 0.05 for cotyledon vs. root.

**Figure 9 ijms-25-00619-f009:**
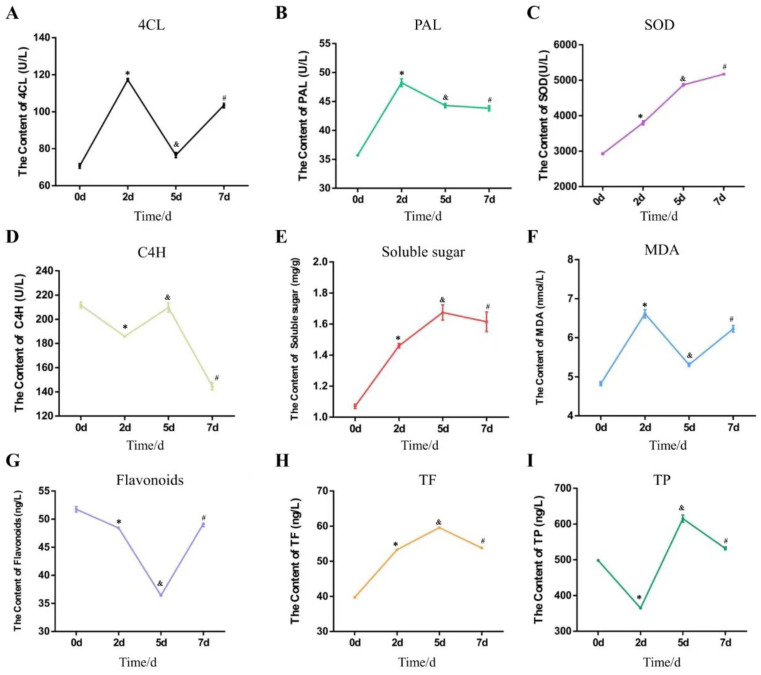
The trends of the contents of 5 metabolites and the activities of 4 key enzymes. The horizontal coordinate is the time of inoculation with *F. oxysporum*. The vertical coordinate is the activity data of enzyme or the content of metabolite. (**A**). The content of 4CL: 4-coenzyme A ligase. (**B**). The content of PAL: phenylalanine ammonia lyase. (**C**). The content of SOD: Superoxide dismutase. (**D**). The content of C4H: cinnamic acid-4-hydroxylase. (**E**). The content of soluble sugars. (**F**). The content of MDA: malondialdehyde. (**G**). The content of plant flavonoids. (**H**). The content of TF: total flavonoids. (**I**). The content of TP: total plant phenolics. Each error bar represents the mean ± standard deviation of 3 repetitions. * means *p*-value < 0.05 for 2 d vs. 0 d. & means *p*-value < 0.05 for 5 d vs. 0 d. # means *p*-value < 0.05 for 7 d vs. 0 d.

**Figure 10 ijms-25-00619-f010:**
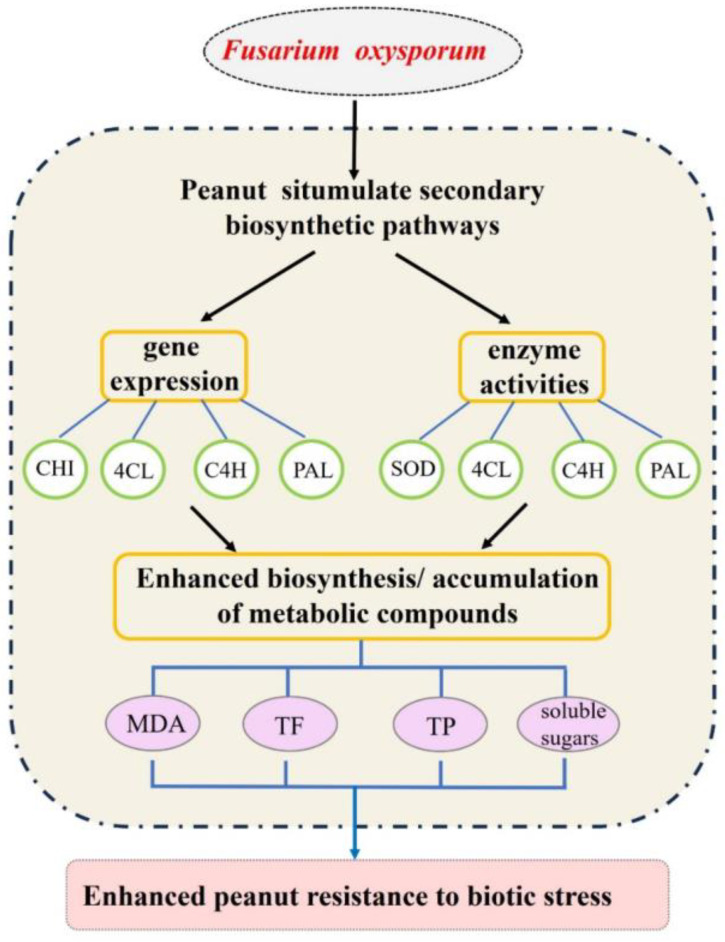
The proposed mechanism of peanut’s response to *F. oxysporum* infection. PAL: phenylalanine ammonia lyase. C4H: cinnamic acid-4-hydroxylase. 4CL: 4-coenzyme A ligase. CHI: chalcone isomerase. MDA: malondialdehyde. TP: total plant phenolics. TF: total flavonoids.

**Table 1 ijms-25-00619-t001:** Transcriptome statistical data.

BMK-ID	Clean Reads	Clean Bases	GC Content	%≥Q30
wm-ck-1	37,451,368	11,190,787,760	44.72	93.80
wm-ck-2	38,209,482	11,378,388,158	45.07	94.1
wm-ck-3	36,157,206	10,806,638,472	44.82	93.71
wm-LG-1	39,863,104	11,892,536,984	44.81	93.49
wm-LG-2	36,858,805	11,005,017,808	44.73	93.93
wm-LG-3	37,493,168	11,208,113,484	44.91	93.98

Note: BMK-ID, sample analysis number of BMK; clean reads, the numbers of pair-end reads in clean data; clean bases, total number of bases; GC content, the percentage of G and C bases in clean data; %≥Q30, the percentage of bases with a mass greater than or equal to 30.

**Table 2 ijms-25-00619-t002:** The key genes in Phenylpropanoid biosynthesis pathway (aip00940).

KEGG Pathway	Fold Enrichment	Protein Accession	Protein Description	Ratio	Sequence Coverage (%)	Mol. Weight (kDa)	MS/MS Counts	Peptides	Lg/Ck Ratio	Lg/Ck *p*-Value
aip00940 Phenylpropanoid biosynthesis up	2.34	C8QSDY.1	Peroxidase superfamily protein	2.085	29.1	37.382	32	7	2.085	7.935 × 10^−5^
	2.34	L4CNUE.1	beta glucosidase 46;	16.294	30.4	66.892	43	13	16.294	0.0002401
	2.34	5H4H17.1	phenylalanine ammonia-lyase 2	8.886	52.2	77.944	74	26	8.886	0.0002387
	2.34	IE3GQ3.1	Peroxidase superfamily protein	3.229	52	31.802	68	10	3.229	3.127 × 10^−6^
	2.34	EEZ4Y8.1	phenylalanine ammonia-lyase 2	284.714	53.4	79.697	101	30	284.714	0.0007626
	2.34	J17R0W.1	aldehyde dehydrogenase family 2 member C4-like	7.863	37.4	55.249	38	17	7.863	0.000939
	2.34	SGZ2CH.1	Cytochrome P450 superfamily protein	4.344	46.6	56.811	75	23	4.344	0.0033236
	2.34	ULUR0X.1	cinnamoyl coa reductase	2.589	56.7	36.14	48	14	2.589	0.0002397
	2.34	I3Z6BS.1	Peroxidase superfamily protein	4.029	60.5	34.435	63	10	4.029	5.461 × 10^−7^
aip00940 Phenylpropanoid biosynthesis down	2.6	333C3Q.1	caffeoyl-CoA 3-O-methyltransferase	0.337	54.7	27.63	28	9	0.337	2.104 × 10^−6^
	2.6	H8P4Q7.2	lysosomal beta glucosidase-like	0.353	22.5	74.236	34	10	0.353	1.683 × 10^−5^
	2.6	PX7TTT.1	lysosomal beta glucosidase-like	0.474	27.9	67.094	19	12	0.474	0.0086181
	2.6	VFYD46.1	Peroxidase superfamily protein	0.435	45.7	35.785	33	12	0.435	0.0039972
	2.6	F20QAX.1	Peroxidase superfamily protein	0.47	48.7	34.222	24	9	0.47	0.0108624
	2.6	7BH383.1	Alkyl hydroperoxide reductase	0.353	43	27.123	42	8	0.353	6.402 × 10^−7^
	2.6	0XM443.1	beta glucosidase 43	0.486	45.2	58.95	79	16	0.486	7.419 × 10^−7^

**Table 3 ijms-25-00619-t003:** The top 10 DEPs in up–up and down–down regulatory type.

Transcription ID	Protein Accession	Protein Description	Ratio	*p*-Value	log2FC x	FDR	log2FC y	Type
LMT3MR	LMT3MR.1	polyphenol oxidase A1	3.138	2.46 × 10^−5^	1.649845	1.02 × 10^−273^	7.164713	Up–up
2J0KXT	2J0KXT.1	Chitinase family protein	6.634	1.95 × 10^−9^	2.729879	5.37 × 10^−225^	5.913499	Up–up
4T2HBI	4T2HBI.1	disease-resistance response protein	62.168	3.18 × 10^−7^	5.9581	3.02 × 10^−78^	5.450114	Up–up
EEZ4Y8	EEZ4Y8.1	phenylalanine ammonia-lyase 2	284.714	7.63 × 10^−3^	8.15337	3.84 × 10^−131^	5.229844	Up–up
GX4Q6M	GX4Q6M.1	disease-resistance response protein	61.5	2.19 × 10^−5^	5.942515	6.37 × 10^−18^	5.18639	Up–up
Y9G6RS	Y9G6RS.1	isoflavone reductase homolog	75.923	5.27 × 10^−8^	6.246465	4.80 × 10^−183^	4.846734	Up–up
JSZ7GP	JSZ7GP.1	isoflavone reductase-like	74.949	9.36 × 10^−8^	6.227837	5.28 × 10^−202^	4.817667	Up–up
0BML12	0BML12.1	aldo/keto reductase family oxidoreductase	132.356	0.00548	7.04828	4.90 × 10^−196^	4.746672	Up–up
X2DMBM	X2DMBM.2	4-hydroxy-3-methylbut-2-enyl diphosphate synthase	3.175	9.69 × 10^−8^	1.666757	7.38 × 10^−194^	4.685075	Up–up
1WY37S	1WY37S.1	Cytochrome P450 superfamily protein	73.074	0.003817	6.191286	5.23 × 10^−171^	4.435238	Up–up
L0R0IL	L0R0IL.1	Non-specific lipid-transfer protein	0.484	0.006977	−1.04692	1.35 × 10^−19^	−5.17955	Down–down
YZ06AV	YZ06AV.1	chlorophyll A/B binding protein 1	0.271	3.59 × 10^−5^	−1.88364	1.02 × 10^−147^	−3.95799	Down–down
P96X61	P96X61.2	fructose-bisphosphate aldolase 2	0.434	0.000585	−1.20423	9.48 × 10^−63^	−3.36928	Down–down
Y7HUGW	Y7HUGW.1	photosystem II oxygen-evolving enhancer protein	0.103	0.000861	−3.27928	1.85 × 10^−56^	−3.07236	Down–down
IYE9TT	IYE9TT.1	L-type lectin-domain containing receptor kinase IX.1-like	0.294	2.25 × 10^−5^	−1.76611	1.26 × 10^−7^	−2.95814	Down–down
TAR5IY	TAR5IY.1	receptor lectin kinase	0.368	1.33 × 10^−6^	−1.44222	1.85 × 10^−25^	−2.93076	Down–down
VFYD46	VFYD46.1	Peroxidase superfamily protein	0.435	3.997 × 10^−3^	−1.20091	6.67 × 10^−28^	−2.59009	Down–down
FN6RIW	FN6RIW.1	glyceraldehyde-3-phosphate dehydrogenase C2	0.324	4.53 × 10^−6^	−1.62593	1.95 × 10^−54^	−2.58726	Down–down
GJ4Q3S	GJ4Q3S.1	MLP-like protein 43	0.279	2.55 × 10^−6^	−1.84166	2.25 × 10^−20^	−2.55715	Down–down
KSV0XM	KSV0XM.1	ATP synthase gamma chain 1 family protein	0.364	1.51 × 10^−5^	−1.45799	3.54 × 10^−18^	−2.50991	Down–down

**Table 4 ijms-25-00619-t004:** The primers for qRT-PCR.

Name of Genes	Forward Primer (5′–3′)	Reverse Primer (5′–3′)
Ah1WY37S.1	CGGAAAGCCCCTCAAGGGTA	TGTTGCGGTGGACCTAGCAA
AhLGAM8W.1	TCCCACCAGGCTCTACTGTCT	ACACTGCCTCTGAAAGTGCCT
Ah0BML12.1	GGGATTGCGGATTCTCATGGC	GTCGTAGCTCTTTGGCACAGC
AhSGZ2CH.1	CTTGGACCAGGCCACCAAGTA	GGTTCATGTGTGGGACGAGGA
AhULUR0X.1	CTGCCTCGAAAGTTCCCTCCA	AACACTTGGTGGGAACAGGGT
Ah333C3Q.1	TGGAGGGTTGATCGGCTATGA	GCAAGGTGCTTGTTGAGTTCC
AhJB63H4.1	TTGGGGGATTGATTGGCTACG	CATGCGCGAGGTACTTGTTGA
Ah5H4H17.1	GCCAAGTTGCCAAGAGGACAC	AGGGTATGTAGCACTGCAAGGA
Ah1KSV8R.1	TCAGGGTTATGGCTTGACGGA	GGGTCGACAATCTTAGCCTGG
AhP96X61.2	CCAGCAAGGTGCTCGTTTCG	GGCTGCTTCCTTAACTGCAAGAG
AhP6MJUK.1	GCCCCAATGATGAGTGCCTTG	CTTCCGGGTTGGCACACATTC
Actin11	TTGGAATGGGTCAGAAGGATGC	AGTGGTGCCTCAGTAAGAAGC

## Data Availability

The sequences data from the six samples obtained are available in the NCBI sequence read archive in Bioproject (PRJNA1020986, TaxID: 226205) with the numbers SAMN37538777, SAMN37538791, SAMN37538792, SAMN37538806, SAMN37538876, and SAMN37538893 for samples wm-ck-1, wm-ck-2, wm-ck-3, wm-LG-1, wm-LG-2, and wm-LG-3, respectively.

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
