# Peer review of "Transcriptomic–Proteomic Analysis Revealed the Regulatory Mechanism of Peanut in Response to Fusarium oxysporum"

_ijms, 2024, doi:10.3390/ijms25010619_

Round 1
Reviewer 1 Report
Comments and Suggestions for Authors
In the MS entitled- Transcriptomic-proteomic analysis revealed the regulatory mechanism of peanut in response to Fusarium oxysporum presented by Wang et al. a very descriptive work providing insights into the regulatory mechanism of peanut in response to Fusarium oxysporum is given. The manuscript is overall well written and understandable, although with some weaknesses in the few parts. Most of the experiments were well performed with proper controls. However, there can be some revisions in the MS, that can be improved for better a better understanding of the reader. Overall, in my opinion, it was a very interesting manuscript to read and could be much more improved by adding some of the following minor suggestions.
1. Authors mentioned, transcriptome and proteome profiling were combined to provide new insights into the molecular mechanisms of peanut - Pls clarify somewhere in the MS and mention Cleary, how the proteome sequencing and transcriptome data/sequencing in your study is corelated?
2. Authors talk about the foundation of biological and chemical analysis for the breeding of resistant varieties, how? Pls mention how breeding would be helpful in peanut against Fusarium oxysporum.
3. Abstract is too technical, improve it for wider reader applicability.
4. Improve discussion part- “Here, our works showed similar result” – make it clear, what does the authors want to say?
5. Improve English in the MS.
Reviewer 2 Report
Comments and Suggestions for Authors
The manuscript entitled "Transcriptomic-proteomic analysis revealed the regulatory mechanism of peanut in response to Fusarium oxysporum"from Wang et al. is an interesting study and research in this field it is needed. The manuscript presents interesting results and strategies to understand Peanut Fusarium rot. In the point of view, this manuscript should be improved for language. Revise the scientific name very well specially in the abstract. In addition, about the novelty, there were a lot of published papers that works with omics. Also, it is important to consider that even with good results presented; the authors should state about the potential use of these technologies in the field, this mean, what about the scale up processes? What about the costs of this system.
And please add the table of primers for RT validation and supplementary data.
Reviewer 3 Report
Comments and Suggestions for Authors
This manuscript explores peanut Fusarium rot, a prevalent issue in Chinese peanut production, seeking to elucidate the regulatory mechanisms of peanut in response to Fusarium oxysporum. Transcriptome and proteome profiling offer new insights into the molecular mechanisms underlying peanut responses post-infection; however, several critical remarks can be made to enhance the manuscript clarity and accuracy.
The abstract requires some refinement to address a few grammar issues and enhance its overall clarity:
Line 12, 15, 29: At first introduce the abbreviation of Fusarium oxysporum and KEGG pathways, and subsequently, use them in the text.
Correct grammatical errors in the abstract, e.g. "after F. oxysporums infected" (line 14); "different express genes" (lines 14 and 15); "different express proteins" (line 15); "after F. oxysporums infected" (line 19), etc.
Avoid redundant use of "significant" (lines 11 and 12) and "provide" (lines 22 and 23).
Line 39: The authors discuss immune response; however, the examples presented do not align with plant immunity. Instead, the focus shifts to other topics, such as proteins associated with the photosynthetic apparatus and energy metabolism.
Line 45: Italicize F. oxysporums
Lines 78-87: The description of the pathogenic phenotype is vague. It would be beneficial to provide more precise details on the observed characteristics, such as size, color or distribution of disease spots. The text also mentions the identification of peanut germplasm resources for disease resistance but provides limited details on the criteria used for identification.
Lines 97-100: The repetition of "7 d after inoculation" in the sentence may cause confusion. The authors likely used it to emphasize the specific time point when the transcriptomic analysis was conducted for the control and the samples after inoculation with F. oxysporum. However, this repetition could be perceived as unnecessary.
Lines 134-146: Some sentences are complex and may be challenging for readers to follow. Simplifying the language and sentence structure can enhance clarity and accessibility.
Line 146: The term "pathogenicity" is used without clarification. It would be helpful to specify whether it refers to the pathogen ability to cause disease or the plant resistance to pathogen infection.
Line 214-217: I recommend rephrasing the sentence to enhance readability. Consider the following revision: "This implies that in response to pathogen infection, peanuts can activate the expression of key genes linked to plant disease resistance and secondary metabolism pathways. Moreover, pathogen invasion has the potential to disrupt or obstruct the photosynthesis pathway in plants, thereby enhancing the success of their pathogenicity".
Lines 233-240: There is some repetition of phrases, such as "levels of expression". Consider varying the language to avoid redundancy.
Lines 257-258: There are a few instances where the use of commas can be confusing. For example, the phrase "The levels of expression of these genes vary in different peanut tissues might play roles in various tissue sites" could benefit from rephrasing.
Lines 264-271: While the text mentions changes in enzyme activities, it lacks specific quantitative data. Adding numerical values or percentage changes would provide a clearer picture of the observed effects.
Line 266: The use of “sickle” is unclear; it might be a typo.
In general, the Discussion section lacks depth and some statements are overly broad. More thorough exploration and interpretation of the findings, especially in the context of the current understanding of plant-pathogen interactions and how these findings align or differ from existing research on plant-Fusarium interactions, would enhance the manuscript value.
Lines 292-296: Revise the sentences for clarity, as it is unclear what you intend to convey.
Line 300: There is a mistake in the phrase "In particularly". The correct expression is "In particular" or "Particularly".
Lines 308-310: Adjust the tense in the sentence from present perfect to past since you are discussing your results.
Line 327: Instead of "were improved after infected" use "were improved after infection".
Lines 339-340: Clarify the sentence "But, the activities of C4H decreased was different result with the other report".
Figure 9 is devoid of statistical analysis, while in Figure 7 and Figure 8, error/deviation bars are displayed without accompanying information about their significance or interpretation.
The Material and Methods section lacks information regarding the statistical analysis.
In summary, the manuscript appears to be scientifically sound, and the experimental design seems suitable for testing the hypothesis. The figures and scheme effectively depict the data, and the conclusions align well with the data presented. However, some data lack proper statistical analysis.
Comments on the Quality of English LanguageThe manuscript requires thorough revision to improve its language and grammar.
Round 2
Reviewer 2 Report
Comments and Suggestions for Authors
Accepted in the present form